# Single Wavelengths of LED Light Supplement Promote the Biosynthesis of Major Cyclic Monoterpenes in Japanese Mint

**DOI:** 10.3390/plants10071420

**Published:** 2021-07-12

**Authors:** Takahiro Ueda, Miki Murata, Ken Yokawa

**Affiliations:** Faculty of Engineering, Kitami Institute of Technology, Hokkaido 090-8507, Japan; t.ueda7991@gmail.com (T.U.); muratamk@mail.kitami-it.ac.jp (M.M.)

**Keywords:** mint, monoterpenes, solid phase microextraction (SPME), hydroponics, LED supplement

## Abstract

Environmental light conditions influence the biosynthesis of monoterpenes in the mint plant. Cyclic terpenes, such as menthol, menthone, pulegone, and menthofuran, are major odor components synthesized in mint leaves. However, it is unclear how light for cultivation affects the contents of these compounds. Artificial lighting using light-emitting diodes (LEDs) for plant cultivation has the advantage of preferential wavelength control. Here, we monitored monoterpene contents in hydroponically cultivated Japanese mint leaves under blue, red, or far-red wavelengths of LED light supplements. Volatile cyclic monoterpenes, pulegone, menthone, menthol, and menthofuran were quantified using the head-space solid phase microextraction method. As a result, all light wavelengths promoted the biosynthesis of the compounds. Remarkably, two weeks of blue-light supplement increased all compounds: pulegone (362% increase compared to the control), menthofuran (285%), menthone (223%), and menthol (389%). Red light slightly promoted pulegone (256%), menthofuran (178%), and menthol (197%). Interestingly, the accumulation of menthone (229%) or menthofuran (339%) was observed with far-red light treatment. The quantification of glandular trichomes density revealed that no increase under light supplement was confirmed. Blue light treatment even suppressed the glandular trichome formation. No promotion of photosynthesis was observed by pulse-amplitude-modulation (PAM) fluorometry. The present result indicates that light supplements directly promoted the biosynthetic pathways of cyclic monoterpenes.

## 1. Introduction

Secondary metabolites are chemical products enzymatically converted from primary metabolites in plants. Plants use these compounds to adapt to their environment, for example, defense against pathogens or insect attacks, or other stresses. In human history, secondary metabolites have also been an essential source of medicines [1]. Although the development of modern chemistry enables us to synthesize a broad range of chemical compounds, many plant secondary metabolites, especially terpenes, are still considered important pharmaceutical materials. This is because it is convenient to use plant-derived compounds as synthetic starting materials to obtain the desired molecules. Thus, significant efforts have been made to acquire valuable secondary metabolites [2,3].

A glandular trichome (GT) is a plant-specific storage organ distributed in the aerial part of the plant body. In mint plants, GTs play an essential role as a tiny cell factory to synthesize and accumulate secondary metabolites. In terms of biotechnological or pharmacological interests, the number of studies on plant GTs has increased recently [4,5]. Transcriptomic analysis was conducted to elucidate the regulation of GT-specific terpene biosynthesis, for example, in spearmint [6] and *Artemisia* plants [7]. An essential oil generated catalytically in GTs is susceptible to environmental fluctuations or stresses [8]. Particularly, an increase in the terpene contents after various abiotic stress treatments in medicinal plants was confirmed [9].

Mint plants belonging to the *Mentha* genus are the most famous herbs that have been used in the past. A major terpene produced by the plant is menthol. Menthol causes a sensation of coolness through the direct reaction with the transient receptor potential melastatin 8 (TRPM8) channel [10]. Menthol-containing essential oil is used extensively for many purposes. Menthol has many biological actions, and its antifungal activity is a well-known function [11]. In addition to menthol, mint plants produce cyclic monoterpenes, for example, pulegone, menthone (the intermediates in menthol biosynthesis), and menthofuran (Figure 1). The balance of these contents features an odor of the mint leaves, and the growth environment influences the biosynthesis of the compounds.

Light is one of the physical factors necessary for plant growth, development, and metabolism. Many studies have focused on light and changes in secondary metabolites, such as anthocyanins, carotenoids, and flavanols, under the control of photoreceptors [12]. The light effect on monoterpene metabolism of mint plants has long been studied [13]. It was assessed that a short-day photoperiod treatment for three mint species significantly increased the oil content [14]. In a controlled light environment using light-emitting diodes (LED), red LED was highly effective in increasing the oil content in *M. piperita*; blue and white LEDs were also effective [15]. However, it is still unclear which wavelength of light affects both the contents and composition of monoterpenes produced by mint plants.

Here, we analyzed the contents of four major cyclic monoterpenes in a Japanese mint plant cultivated under single-wavelength supplementation of blue, red, or far-red LED light. The number of GTs on the growing mint leaves was counted and compared between the treatments. The possible physiological mechanism of light supplementation on terpene biosynthesis was also discussed.

## 2. Materials and Methods

### 2.1. Plant Preparation and Growth Conditions

A rootstock of the major Japanese mint cultivar “HOKKAI JM 23” (*Mentha canadensis* L.) was obtained from the National Agriculture and Food Research Organization Genebank, Tsukuba, Japan (collection ID: JP176265). The plants were first recovered from the rootstock and grown in commercial soil. Fourteen young shoots were harvested from a well-grown mint plant by stem cutting at the position below the fourth leaf. The cuttings were incubated until adventitious roots were generated in distilled water for nine days at room temperature under a 16 h/8 h light/dark cycle at 23 °C. The young plants were then placed in a commercially available hydroponic cultivating system, as shown in Figure 2A (Green Farm, U-ing, Osaka, Japan). The box-shaped system was equipped with programmable white LED lighting, a ventilator, and automatic water flow/aeration. For acclimation to the new environment, the plants were pre-grown for one week with distilled water. At the beginning of the experiment, the hydroponic water tank was filled with 4 L of standard Hoagland’s cultivating solution (2.5 mM KNO_3_, 1.25 mM Ca(NO_3_)_2_·4H_2_O, 0.5 mM NH_4_NO_3_, 1 mM MgSo_4_·7H_2_O, 0.25 mM KH_2_PO_4_, 25 mM NaFe(Ⅲ)EDTA, 23 mM H_3_BO_3_, 4.55 mM MnCl_2_·4H_2_O, 0.39 mM ZnSO_4_·7H_2_O, 0.1 mM CuSO_4_·5H_2_O, and 0.25 mM Na_2_MoO_4_·2H_2_O; FUJIFILM Wako Chemicals, Osaka, Japan), and the pH of the solution was measured daily using a portable pH meter (LAQUAtwin pH-33B, HORIBA, Kyoto, Japan). The pH was adjusted to 6.2 using 1 M 2-morpholinoethanesulfonic acid (Dojindo, Kumamoto, Japan) buffer, and 5 M NaOH was used every two days [16].

### 2.2. LED Lighting Conditions

The photosynthetic photon flux density (PPFD) of all light sources used in the study was measured using a Light Analyzer LA-105 (NK Systems, Tokyo, Japan). In the cultivating system, the average PPFD value of white light (WL) LED at the foliar position was at 161 µmol/m^2^/s. An arrayed LED source of blue light (BL), red light (RL), or far-red (FR) supplements was placed on the ceiling of the cultivating box. All light spectra are shown in Figure 2B. PPFD values of BL, RL, and FR were at 6.7, 7.1, and 3.7 µmol/m^2^/s, respectively. The photocycle of the basal WL was 16 h light/8 h dark. The daily light supplement of each wavelength began at the same time when the WL was on and lasted 6 h. In the hydroponic culture system used in the study, 16 young mint plants were simultaneously cultivated and treated with LED light for each independent experiment. The duration of the experiment of each light supplement was two weeks.

### 2.3. HS-SPME and GC–MS Analysis for Cyclic Monoterpenes

After two weeks of light treatment, the second and third leaves (total four leaves) were harvested by cutting the petiole and weighed. The first leaves newly emerged during the 2-week LED treatment, and the size of the leaves was small and immature. To observe the effect of the LED supplements on expanded leaves, we decided to use the second and third leaves for further analyses. The analysis of a volatile compound with the head-space solid phase microextraction (HS-SPME) method was adapted from a previous report [17]. The leaves were then frozen with liquid nitrogen and ground finely with a pre-cooled pestle and mortar. The ground powder was put into 2 mL of 2 M CaCl_2_·H_2_O solution in a head-space 20-mL glass vial (Supelco/Sigma-Aldrich, Tokyo, Japan) to prevent unnecessary enzymatic reactions. The tightly sealed vials were incubated in a hot bath at a temperature of 40 °C for 5 min. An SPME fiber coated with 60-µm-thick PDMS/DVB (polydimethylsiloxane/divinylbenzene, Supleco) was inserted into the head-space through the vial septum, and the volatile released from the warmed solution was absorbed. The fiber was removed from the vial and inserted quickly into the inlet of a gas chromatography–mass spectroscopy (GC–MS) system (GC-17A, Shimadzu, Kyoto, Japan) with a polyethylene glycol column (60 m long, 0.25 mm internal diameter, 0.25 µm film thickness; GL Sciences, Tokyo, Japan). The temperature of the inlet was held at 180 °C with a 9:1 split. Helium was used as the carrier gas at a 2 mL/min flow rate, 245.9 kPa. The starting oven temperature was 100 °C, followed by an 8 °C/min ramp until 180 °C was reached and held for 2 min. The interface to the mass spectrometer was held at 250 °C. The mass spectrometer was run in the scan mode with electron impact ionization at 1 keV, from *m*/*z* 40 to *m*/*z* 300. Compounds were identified using the National Institute of Standard and Technology Mass Spectral Database (NIST12). The GC–MS analyses of terpenes were conducted independently six times in different LED lightning treatments.

### 2.4. Preparation of Standards

Standard solutions of pure chemicals of menthol and menthone (FUJIFILM Wako Chemicals, Osaka, Japan), pulegone, and menthofuran (Sigma-Aldrich, Tokyo, Japan) were dissolved in pure methanol. The dilution series of each solution was prepared with methanol. A volume of 2 µL of the standard solution was added to the glass vial with 2 mL CaCl_2_, and the compound was measured using GC–MS with the same method described above. Three to four concentrations of each solution were used to make standard curves. Peak areas of each compound obtained both from samples and standards were used for calculations.

### 2.5. Quantification of GTs

All experiments were conducted using plant samples grown under different light supplements for two weeks. At the bottom, middle, or top of the leaf on both the abaxial and adaxial sides, six locations were imaged using a stereomicroscope (WRAYMER 820T, WRAYMER, Osaka, Japan). In the same manner, the number of GTs on the second and third leaves was compared. An area of 4 mm^2^ was chosen randomly from each picture, and the total number of GTs was counted. The data were analyzed independently from sixteen plants as biological replications.

### 2.6. Chlorophyll Quantification

Chlorophyll was extracted and quantified according to a previous method [18]. The chlorophyll and anthocyanin extractions were made using leaves different from the ones used for the terpene quantifications. The leaf position (e.g., lighting) alters the chlorophyll contents. Therefore, to minimize the errors among the analyses, we carefully harvested and chose the second and third leaves with similar sizes for terpene or chlorophyll measurement. Briefly, weighed leaf samples were placed into 2 mL of pure N,N-dimethylformamide (DMF; FUJIFILM Wako Chemicals, Osaka, Japan) at 4 °C in darkness overnight. The absorbance values at 647 and 664 nm of the aliquot of the extracted solution were measured using a spectrophotometer (U-5100, Hitachi, Tokyo, Japan). DMF was used as the blank control. The total amount of chlorophyll was calculated based on the following equation [18]: Chl a + b = 17.67 ∗ A_647_ + 7.12 ∗ A_664_.

### 2.7. Anthocyanin Quantification

For anthocyanin extraction and quantification, a method reported previously was adapted to this study [19,20]. Weighed leaf samples were ground with liquid nitrogen with a pre-cooled pestle and mortar. The powder was put into 2 mL of 1% HCl–methanol and incubated overnight at 4 °C in darkness. A volume of 300 µL of the aliquot was collected into a new 1.5 mL tube, and 200 µL of distilled water and 500 µL of chloroform were added. After mixing, the tube was centrifuged for 5 min at 13,000× *g* at 4 °C. A volume of 400 µL of the aliquot was collected carefully into a new tube, and 400 µL of 60% methanol + 1% HCl solution was added. The absorbance of the solution containing anthocyanin was measured at 530 nm and 657 nm.

### 2.8. Measurement of Photosynthetic Efficiency

The efficiency of photosynthesis was non-invasively monitored using a pulse-amplitude-modulation (PAM) chlorophyll fluorometer in each plant grown under different light wavelengths. Plants were placed in a dark environment for 30 min before the PAM measurement to obtain the maximum quantum efficiency of PSII (Fv/Fm). Then, PAM measurements were conducted under dim light using Junior-PAM (WALZ, Effeltrich, Germany).

### 2.9. Statistical Analyses

All numerical data were analyzed using Tukey’s honestly significant difference (HSD) for parametric analysis and the Mann–Whitney *U* test for non-parametric analyses with R software version 4.0.2. (https://www.r-project.org/ accessed on 23 July 2020). Differences were considered significant when *p*-values were smaller than 0.05.

## 3. Results and Discussion

### 3.1. Promotion of Terpene Biosynthesis by Light Supplements

After two weeks of light supplementation, the increase of the cyclic monoterpene content in the treated leaves was observed. Especially, BL increased all four monoterpenes (Figure 3): pulegone (362% increase compared to the WL control), menthofuran (285%), menthone (223%), and menthol (389%). RL slightly promoted pulegone (256%), menthofuran (178%), and menthol (197%). These results are consistent with a previous study that showed an increase in fresh weight and essential oils in several mint species (*Mentha piperita*, *Mentha spicata*, etc.) cultivated in a red-blue LED incubator [15]. Another previous report also observed the increase of terpene contents in *Cannabis* plants under RL-BL sub-canopy LED lighting [21]. The precursors of all higher plant monoterpene biosynthesis, isopentenyl diphosphate (IPP) or dimethylallyl pyrophosphate (DMAPP), were proposed to be increased by the RL-BL supplement [21]. It is likely to explain our result showing the increase of cyclic monoterpene with either BL or RL supplement in Japanese mint (Figure 3). Interestingly, the accumulation of menthone (229%) or menthofuran (339%) was observed with FR light treatment. Our result is the first report that showed that FR light treatment increased terpenes in mint plants. In natural conditions, long-day treatment to mint plants showed the accumulation of menthofuran [14]. It indicates that the content of menthofuran is associated with the life cycle of the mint plant. Our result showed that the FR treatment disturbed the plant response to the photoperiod, affecting the metabolism of menthofuran. Furthermore, the reaction to FR indicates that the biosynthesis of these compounds is possibly under the control of phytochromes. In our study, no significant promotion of plant growth was observed during two weeks of LED light supplementation at the fluence rate used. Also, the contents of cyclic monoterpenes shown in Figure 3 were normalized with the weight of fresh leaves. This suggests that the increase of cyclic monoterpenes was not due to the leaf area’s expansion. In many *Lamiaceae* plants, GTs in the aerial part of the plant body influence the biosynthesis of secondary metabolites, including terpenes, and store them in a cavity space surrounded by a cuticle. Here, we hypothesized why the terpenes increased after LED treatments as follows: (1) BL treatment increased the density of GTs, and (2) light stimulated the processes of terpene biosynthesis.

### 3.2. Alteration of GT Density

We next observed whether the light treatment promoted the emergence of GTs. After two weeks of LED light treatment, the second and third leaves from the top of mint cuttings were harvested. The number of GTs in three regions of the leaf (apex, middle and basal regions) was counted separately and averaged. Overall, the density of GTs in the abaxial side of the leaves was slightly higher than that on the adaxial side (Figure 4A,B and Figure 5A,B). On both sides of the second leaf, except the apex region of the abaxial side, the GT density was decreased by the BL treatment (Figure 4A,B). There was a tendency for an increase in GT density by the FR light treatment. On the third leaf, a BL-dependent GT increase was similarly observed, whereas FR showed no promotion of GT emergence (Figure 5A,B). We observed three regions of the second and third leaves to determine whether GT was newly generated as the leaf expands (apex region) or light directly regulated GT density regardless of the leaf regions. Here, we confirmed that environmental factors, including light conditions, could flexibly control GT density. A previous study showed that two light conditions (full solar radiation vs. shade) had no impact on GT density in a medicinal plant, *Ocimum campechianum* [22]. In cultivated tomato plants, a density of type VI leaf GTs increased under high light conditions (approximately 300 µmol/m^2^/s) [23]. Thus, light supplementation is likely to affect the density of GTs. The emergence of GTs is under the regulation of the well-known MYB transcription factor. Overexpression of AaMYB17 in sweet wormwood (*Artemisia annua* L.) increased the density of GTs [24]. Homeodomain-leucine zipper IV transcription factor was also shown to be involved in the control of GT density [25,26]. Our results show that the modulation of GT density by BL and FR is possible because of the regulation of transcription factors by light perception. However, the reduction of GT density with BL treatment was inconsistent with the increase of terpene contents, as shown in Figure 3. This suggests that biological processes of terpene biosynthesis were promoted directly by single wavelengths of LED light supplements.

### 3.3. Photosynthesis Was Not Boosted by LED Light Supplements

In mint plants, monoterpenes are biosynthesized from isopentenyl pyrophosphate and dimethylallyl pyrophosphate, which are provided by the plastidial methylerythritol phosphate (MEP) pathway [27]. The initial compound of the MEP pathway is pyruvic acid, which is supplied from sugars as a photosynthetic by-product. Next, we confirmed whether the photosynthetic activity was the source of monoterpene biosynthesis under light supplementation. We measured chlorophyll contents in the two-week light-treated mint leaves and compared them among the different light wavelengths. Similar to GT density results, BL and FR treatments significantly increased chlorophyll but not anthocyanin (Figure 6A). Anthocyanin biosynthesis was induced by BL treatment to plants. In *Arabidopsis thaliana*, anthocyanin biosynthesis was stimulated by 2.5 W/m^2^ of BL (approximately 11 µmol/m^2^/s) [28], which is higher than the intensity we used in this study (6.7 µmol/m^2^/s). We considered that the increase in the chlorophyll contents induced by either BL or FR supplements could help the biosynthesis of cyclic monoterpenes in GTs. To check the contribution of chlorophyll to photosynthesis, we quantified the quantum yield of photosystem II based on chlorophyll fluorescence.

Contrary to our expectations, both Fv/Fm and the quantum yield showed no change with any wavelength of light treatment (Figure 6B). This suggests that the increase in cyclic monoterpenes shown in Figure 3 was not due to the promotion of photosynthesis by the light supplements. Evans and Terashima reported no correlation between the chlorophyll content in spinach leaves and photosynthetic activity [29]. In *Arabidopsis*, red and far-red wavelengths supplemented with WL enhanced chlorophyll biosynthesis [30]. A BL supplement was also shown to increase the chlorophyll content in seven plant species [31]. In conclusion, chlorophyll generation observed in light-supplementation experiments might be the result of the transient activation of photoreceptors and does not influence photosynthesis and terpene biosynthesis.

## 4. Conclusions

We propose a model based on the results as shown in Figure 7. The background WL illumination was still important for photosynthesis to provide sugars for terpene biosynthesis in GTs. We then speculate that the increase in cyclic monoterpenes in the GTs was probably due to the direct activation of processes of terpene biosynthesis. Studies have revealed that light modulated the biosynthesis of terpenes in many plant species. For example, a RL receptor, phytochrome, was shown to promote monoterpene production in thyme seedlings [32]. In cannabis plants, the content of a meroterpenoid, cannabinoid, was increased by BL treatment with a short photoperiod (12 h light/12 h dark) [33]. This indicates that photoreceptors or related factors, such as transcription factors, might directly or indirectly facilitate the enzymatic processes of terpene biosynthesis in secretory cells in GTs.

Recently, GT research is spotlighted because certain secondary metabolites produced by plants have high pharmaceutical value [4,5]. A lot of chemical compounds for producing medicines still rely on plant-derived starting materials. For example, *Artemisia annua* is recognized as a precious plant resource of artemisinin, an antimalarial agent, stored in GTs. Since a net synthesis of artemisinin has great difficulty and costly, the cultivation of *Artemisia* plants is necessary to obtain the compound. In addition to the plant, many other medicinal plants store valuable second metabolites, including terpenes in GTs. It is known that the contents of the compounds in GTs are dependent on the growing environment [8]. Therefore, a considerable effort is being paid to cultivating medicinal plants in an artificially modified environment to maximize the contents of desired compounds in GTs. Results obtained from Japanese mint as a model of GTs metabolism showed that light, which is one of the critical environmental factors, promotes monoterpene synthesis, and it will be helpful for GT research. Although we first speculated that the BL affects specific terpene biosynthesis, all four monoterpenes (pulegone, menthofuran, menthone, and menthol) were significantly increased. It suggests that the BL might stimulate the biosynthesis and accumulation of terpene precursors such as DMAPP or IPP in GTs (Figure 1 and Figure 7). These compounds are the necessary starting materials for all higher plant monoterpene biosynthesis, including *Artemisia* plants, as mentioned above. Thus, the information on supplement lighting obtained from the Japanese mint study will be helpful to boost the contents of valuable terpene compounds in medicinal plants for industrial cultivating conditions. As no genomic information on the Japanese mint used in the study was provided, genetic verification was not conducted in this study. Therefore, in future research, further details on physiological regulation by the light perception in the Japanese mint need clarification.

## Figures and Tables

**Figure 1 plants-10-01420-f001:**
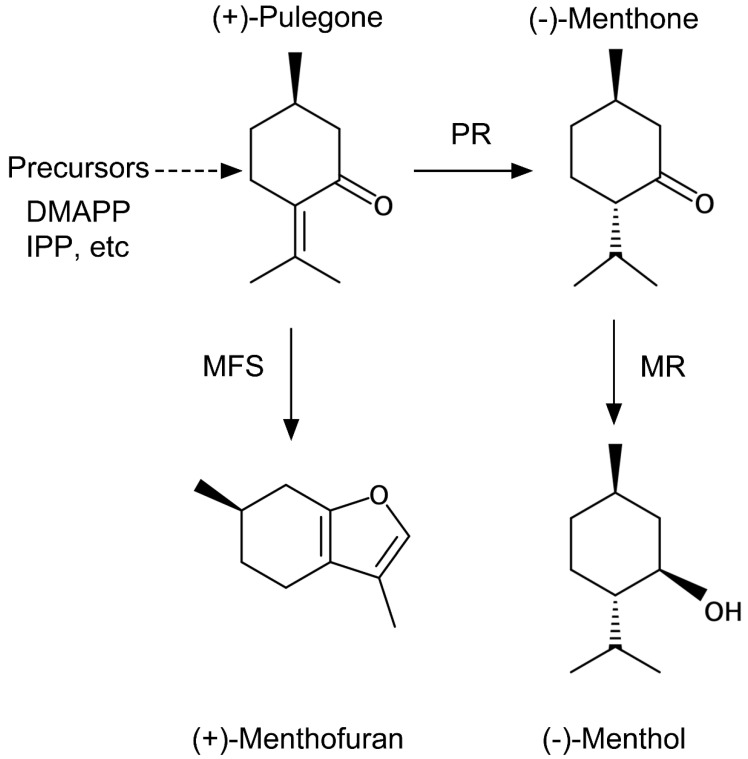
Cyclic monoterpene 12 synthesized in glandular trichomes in Japanese mint. These four molecules are the major components of Japanese mint flavor. PR: pulegone reductase; MR: (−)-menthol reductase; MFS: menthofuran synthase.

**Figure 2 plants-10-01420-f002:**
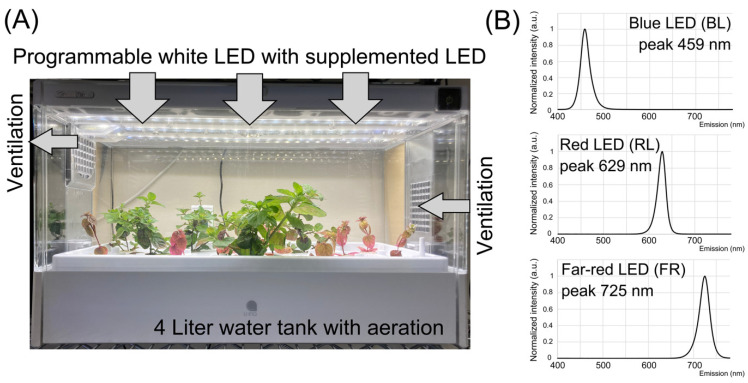
Hydroponic cultivation of Japanese mint and LED supplement. (**A**) A commercially available hydroponic system was modified for the light-supplementation experiment. The system is semi-closed and automatic. (**B**) Three light spectra were emitted from three LED sources used in the study: blue light (BL), red light (RL), and far-red (FR).

**Figure 3 plants-10-01420-f003:**
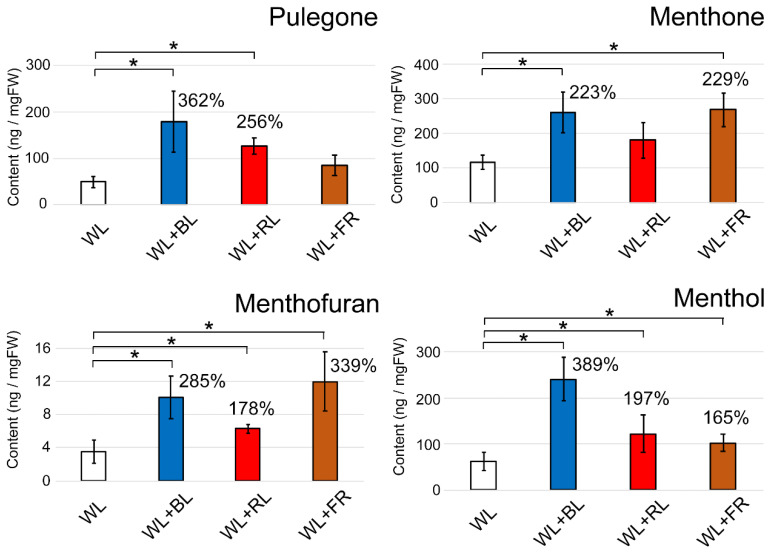
Modulation of the contents of cyclic monoterpenes in two-week light-supplemented Japanese mint leaves. The contents of pulegone, menthone, menthofuran, and menthol are presented. The values in the graphs indicate the increasing rates compared to the WL control. The measurements were conducted independently six times for biological replicates. Error bars indicate standard deviations from the mean. * Asterisks represent a significant difference from the WL control in the Mann–Whitney *U* test (*p* < 0.05).

**Figure 4 plants-10-01420-f004:**
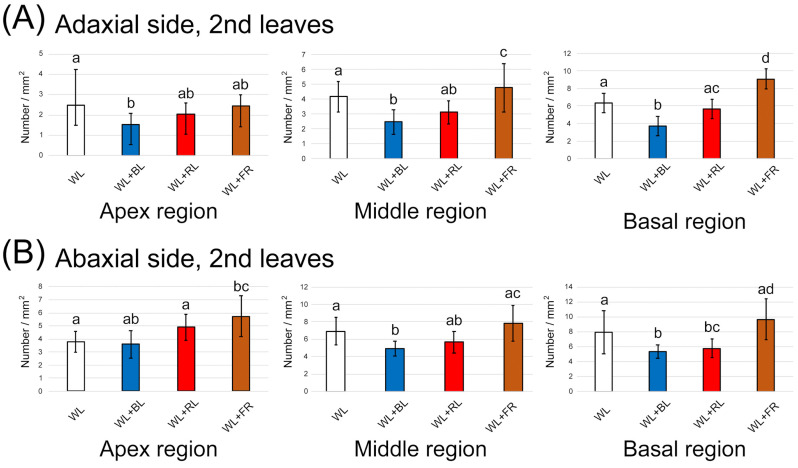
The number of glandular trichomes generated on the second leaf under different light treatments for two weeks. (**A**) Quantification on the adaxial side in the three leaf regions (*n* = 16). (**B**) Quantification on the abaxial side in the three leaf regions (*n* = 16). The data were obtained from four plants in four independent experiments as biological replications. Error bars indicate standard deviations from the mean. Different alphabets indicate significant differences according to Tukey’s HSD test; *p* < 0.05.

**Figure 5 plants-10-01420-f005:**
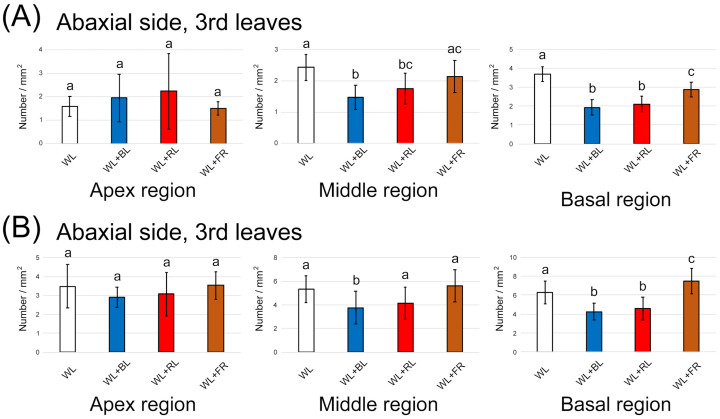
The number of glandular trichomes generated on the third leaf under different light treatments for two weeks. (**A**) Quantification on the adaxial side in the three leaf regions (*n* = 16). (**B**) Quantification on the abaxial side in the three leaf regions (*n* = 16). The data were obtained from four plants in four independent experiments as biological replications. Error bars indicate standard deviations from the mean. Different alphabets indicate significant differences according to Tukey’s HSD test; *p* < 0.05.

**Figure 6 plants-10-01420-f006:**
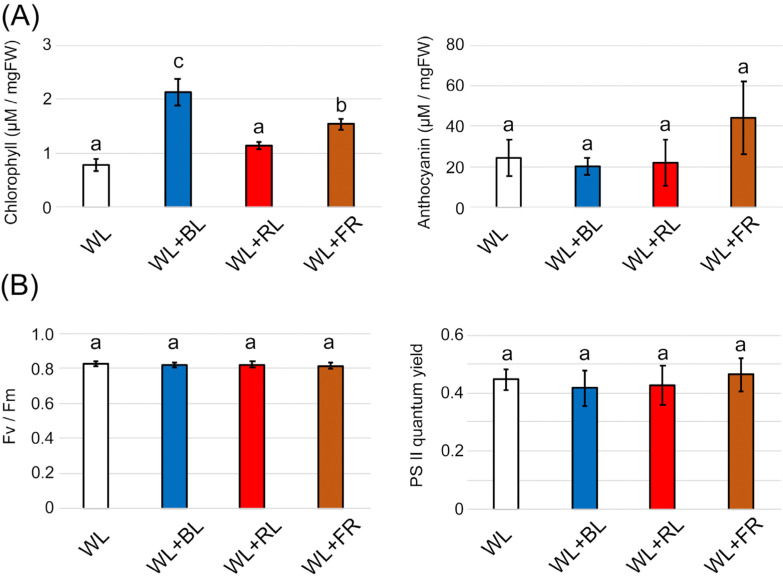
Effects of light supplementation on leaf pigment and photosynthesis in two-week light-supplemented Japanese mint leaves. (**A**) Total chlorophyll (*n* = 4) and anthocyanin contents (*n* = 4) were quantified using a spectrophotometer. (**B**) Photosynthetic parameters, Fv/Fm, and PSII quantum yield on leaves were monitored using a PAM chlorophyll fluorometer (*n* = 8–13). The data were obtained from independent experiments as biological replications. Error bars indicate standard deviations from the mean. Different alphabets indicate significant differences according to Tukey’s HSD test; *p* < 0.05.

**Figure 7 plants-10-01420-f007:**
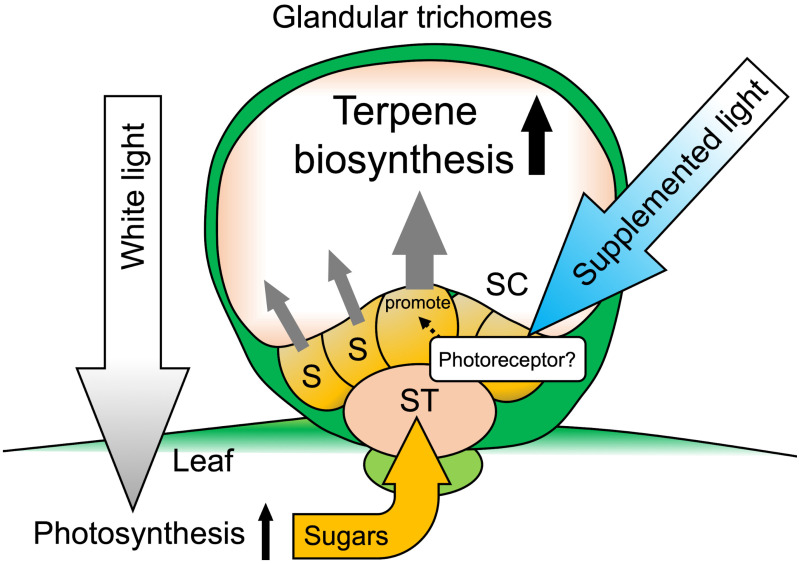
Schematic of glandular trichomes on Japanese mint leaves. Background WL provides energy for photosynthesis and provides a sugar source, which is necessary for terpene biosynthesis in glandular trichomes. LED light supplementation may promote the terpene biosynthetic pathway in glandular trichome cells through photoreceptor activation. S: secretory cells; ST: stalk cells; SC: storage cavity.

## Data Availability

The data that support the findings of this study are available from the corresponding author, K.Y., upon reasonable request.

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
