# Peer review of "Single Wavelengths of LED Light Supplement Promote the Biosynthesis of Major Cyclic Monoterpenes in Japanese Mint"

_plants, 2021, doi:10.3390/plants10071420_

Round 1

Reviewer 1 Report

1.    My concerns are related to checking the manuscript for the English language errors carefully.
2. Restructure the abstract and add more information about the results obtained.
3.    Results are not comprehensively written and can be elaborated with more details.
4.    Discussion can be improved from examples for the literature and more references to relate the results obtained.
5.     Also update and replace old references with recent references
6. Check figures and their ligands, especially Figure 4 and 5 are confusing.
7. Elaborate the discussion section with a correlation of the studies with your work.
8. Restructure and carefully edit the conclusion section, Figure Required?.

Author Response

Response to reviewer #1:

Thank you very much for your valuable comments. With your suggestions, our manuscript was  improved very much. The followings are our replies to your points.

  1.   My concerns are related to checking the manuscript for the English language errors carefully.

We have sent the manuscript to an English editing service, Enago, and checked the errors again throughout the text.

  1. Restructure the abstract and add more information about the results obtained.

Thank you very much for the suggestion. We have reconstructed and improved the abstract to include more detail about the results we obtained.

  1. Results are not comprehensively written and can be elaborated with more details.

We noticed that the lack of descriptions and discussion of the wavelength-dependent increase of monoterpenes. Therefore, we have re-written and added the details focusing on this issue. We also updated figure 3 to indicate the precursor could be the critical target of the light-dependent increase of monoterpene.

  1. Discussion can be improved from examples for the literature and more references to relate the results obtained.

As you pointed out in comment #3, we have improved the texts in the discussion and added new citations to strengthen the discussion part.

  1. Also update and replace old references with recent references

We have updated the list.

  1. Check figures and their ligands, especially Figure 4 and 5 are confusing.

We have checked all the figures. Especially, figure 5 was updated to indicate the actual value of how much percentage the light treatment increased the content of monoterpenes. We also checked all the figure legends and added the biological replications for the statistical analyses. Thank you very much for the suggestion.

  1. Elaborate the discussion section with a correlation of the studies with your work.

Regarding your comment #4, we have improved the discussion part together with the results. In addition to the discussion, we have also modified the conclusion section to elaborate on the perspectives.

  1. Restructure and carefully edit the conclusion section, Figure Required?.

For the conclusion part in the revised manuscript, we have added the descriptions about the advantage of using the Japanese mint plant for the terpene synthesis in glandular trichomes in terms of pharmaceutical study. Your suggestion on the issue was helpful to improve the conclusion, which makes the goal of the manuscript clearer.

Reviewer 2 Report

In this manuscript the authors have investigated the content of monoterpenes in Japanese mint treated with different light wavelength (LED souses). Here following my comments:

  • Both the number of biological and technical replicates have to be clearly indicated in each figure. I assumed that n is the number of independent experiments, however there is no information about the number of biological and technical replicates.
  • Figure 3. I suggest including either fold -change or the percentage of increase of different monoterpenes in plants treated with different LD wave-length vs the control plants.
  • Figure 3. Clearly explain why you have different profiles of monoterpenes (+/- increase) at different light conditions (BL, RF, FR). Is this related on their position in the biosynthetic metabolic pathway (Figure 1)? I strongly suggest discussing these results.
  • Line 114 “After 2 weeks of light treatment, the second and third leaves (total four leaves) were harvested by cutting the petiole and weighed”. Why did you choose the second and third leaves? This point needs to be justified.
  • Are both the chlorophyll content and photosynthesis performances measured using the same leaves used for quantification of monoterpenes? For instance, the chlorophyll content is variable depending on the leaf position. If you correlate the monoterpene profile and chlorophyll content, I assume these analyses have been performed using the same plant material.
  • Considering the pharmaceutical application, I strongly suggest to discuss (more in deep) the potential beneficial effects of increasing monoterpenes in Japanese mint.

Author Response

Response to reviewer #2

Thank you very much for your valuable comments. With your suggestions, our manuscript was  improved very much. The followings are our replies to your points.

  • Both the number of biological and technical replicates have to be clearly indicated in each figure. I assumed that n is the number of independent experiments, however there is no information about the number of biological and technical replicates.

We have modified the description in all figure legends and the materials and methods. As you indicated, the number of “n” means the independent experiments. That means, the number describes biological replicates conducted in different experimental setups (the data were not obtained from the same lightning experiments). The data based on the biological replicates were all used for the statistical analyses.

  • Figure 3. I suggest including either fold -change or the percentage of increase of different monoterpenes in plants treated with different LD wave-length vs the control plants.

We have added the percentage values into figure 3 to indicate the increasing rates compared to the control. In addition, the following sentence was also added to the figure legend.

“The values in the graphs indicate the increasing rates compared to the WL control.”

  • Figure 3. Clearly explain why you have different profiles of monoterpenes (+/- increase) at different light conditions (BL, RF, FR). Is this related on their position in the biosynthetic metabolic pathway (Figure 1)? I strongly suggest discussing these results.

Thank you very much for the suggestion. We have added and improved the discussion about the issue in terms of plant biology. New citations were also added.

Figure 3 was also updated (the names of precursors were added).

  • Line 114 “After 2 weeks of light treatment, the second and third leaves (total four leaves) were harvested by cutting the petiole and weighed”. Why did you choose the second and third leaves? This point needs to be justified.

We added the following sentence to justify why we choose the 2nd and 3rd leaves for the analyses.

“The first leaves newly emerged during the 2-weeks LED treatment, and the size of the leaves was small and immature. Therefore, to observe the effect of the LED supplements on expanded leaves, we decided to use the second and third leaves for further analyses.”

  • Are both the chlorophyll content and photosynthesis performances measured using the same leaves used for quantification of monoterpenes? For instance, the chlorophyll content is variable depending on the leaf position. If you correlate the monoterpene profile and chlorophyll content, I assume these analyses have been performed using the same plant material.

We apologize that the description of the plant materials for chlorophyll and anthocyanin measurements was missing.

The chlorophyll and anthocyanin extractions were made using leaves different from the ones used for the terpene quantifications. We had to grind entire harvested leaves for the terpene analyses because the terpenes' contents were tiny. However, as you are concerned, we have also thought that the leaf position (lighting) should have been a critical factor for the chlorophyll contents. In the hydroponic culture system used in the study, 16 young mint plants were simultaneously cultivated and treated with LED light for each independent experiment. We carefully harvested and chose the second and third leaves with similar sizes for terpene or chlorophyll measurement to minimize the errors in the analyses. We have added the following sentences to the Materials & Methods section.

(LED lighting conditions)

“In the hydroponic culture system used in the study, 16 young mint plants were simultaneously cultivated and treated with LED light for each independent experiment.”

(Chlorophyll quantification)

“The chlorophyll and anthocyanin extractions were made using leaves different from those used for the terpene quantifications. The leaf position (e.g., lighting) alters the chlorophyll contents. To minimize the errors among the analyses, we carefully harvested and chose the second and third leaves with similar size for terpene or chlorophyll measurement.”

  • Considering the pharmaceutical application, I strongly suggest to discuss (more in deep) the potential beneficial effects of increasing monoterpenes in Japanese mint.

We appreciate your suggestion to deepen the discussion about this point.

We have added the following sentences into the discussion part.

“A lot of chemical compounds for producing medicines still rely on plant-derived starting materials. For example, Artemisia annua is recognized as a precious resource of artemisinin, an antimalarial agent. Since a net synthesis of artemisinin has great difficulty and costly, the cultivation of Artemisia plants is necessary to obtain artemisinin accumulated in GT. In addition to the plant, many other medicinal plants store valuable second metabolites, including terpenes in GT. It is known that the contents of the compounds in GT are dependent on the growing environment (Selmar and Kleinwächter, 2013). Therefore, a big effort is being paid to cultivate medicinal plants in the artificially -modified environment to maximize the contents of desired compounds in GT. Results obtained from Japanese mint as a model of GT metabolism showed that light, which is one of the important environmental factors, promotes monoterpene synthesis, and it will be helpful for GT research.”

Round 2

Reviewer 2 Report

The manuscript is focused on optimizing light condition to promote accumulation of monoterpenes. In the conclusion/discussion it is still not clear what the advantage/effect of having this increase amount (at the % described in the manuscript) of pulegone, mentho-furan, menthone, and menthol in mint, in term of industrial, pharmaceutical….. application.

Author Response

Thank you very much for your careful reading especially in the conclusion again.

To make it clear, we have reconstructed the final paragraph in the conclusion and added the following sentence. How our results can be generalize as light(LED)-supported cultivating condition for other plants producing valuable terpenes in industry or pharmaceutical purposes.

"Although we first speculated that the BL affects specific terpene biosynthesis, all four monoterpenes (pulegone, menthofuran, menthone, and menthol) were significantly increased. It suggests that the BL might stimulate the biosynthesis and accumulation of terpene precursors such as DMAPP or IPP in GT (Fig.1 and 7). These compounds are the necessary starting materials for all higher plant monoterpene biosynthesis, including Artemisia plants, as mentioned above. Thus, the information on supplement lighting obtained from the Japanese mint study will be helpful to boost the contents of valuable terpene compounds in medicinal plants for industrial cultivating conditions. "